# Examining the Social Costs of Carbon Emissions and the Ecosystem Service Value in Island Ecosystems: An Analysis of the Zhoushan Archipelago

Qian Zhou [1], Feng Gui [1], Benxuan Zhao [2], Jingyi Liu [3], Huiwen Cai [1], Kaida Xu [4] and Sheng Zhao [1,*]

1   Marine Science and Technology College, Zhejiang Ocean University, Zhoushan 316022, China
2   Zhejiang Zhonglan Environmental Technology Co., Ltd., Wenzhou 325000, China
3   National Engineering Research Center for Marine Aquaculture, Zhejiang Ocean University, Zhoushan 316022, China
4   Zhejiang Province Key Laboratory of Mariculture and Enhancement, Zhejiang Marine Fisheries Research Institute, Zhoushan 316022, China
*   Correspondence: zhaosh@zjou.edu.cn; Tel.: +86-134-5406-9742

**Abstract:** Assessments of the ecosystem service value (ESV) and the social cost of carbon (SCC) inform national and government management decisions in the areas of human well-being and climate change mitigation and adaptation, respectively. Studying the correlation between the two provides a theoretical basis for low-carbon and high-quality regional development, achieving economic decarbonization, and improving human well-being. In this study, we take Zhoushan Archipelago as a case study, consider the ESV and SCC in Zhoushan Archipelago during the period 2010–2020, analyze their spatial development characteristics, and analyze the correlation between the two in time and space. The findings indicate that, with only a 1.5% change, the overall ESV in the Zhoushan Archipelago fell between 2010 and 2020. Conversely, there was a $1604.01 \times 10^4$ t increase in net carbon emissions and a quick 2452% increase in SCC. During the study period, a substantial positive association was found between ESV and SCC in the Zhoushan Archipelago, according to the global spatial correlation analysis of the two variables. It passed the test for *p*-value. This study presents a new potential way to solve the environmental and economic difficulties caused by climate change by providing a mechanism for quantitatively assessing the environment from the perspective of monetary worth. In order to improve the ecological security pattern and ease the burden of regional carbon emissions, it is vital to make use of regional advantages, maintain forests, and develop blue-carbon resources such as mudflats. It is a good idea to cooperate regionally with nearby metropolitan agglomerations. The study's findings are crucial for advancing sustainable development planning in the Zhoushan Archipelago, both theoretically and practically.

**Keywords:** ecosystem service value; social cost of carbon; Zhoushan Archipelago; spatial correlation

## 1. Introduction

Mitigating and adapting to climate change, conserving biodiversity, and ensuring human well-being are three central challenges facing humanity today, and these issues are often treated separately when, in fact, they are deeply intertwined, with many of the same drivers. Finding an integrated approach that can reduce trade-offs and promote synergies between the SDGs is a focus of current and future research [1]. Adaptation to climate change and sustainable development are themes of international relevance [2]. Sustainable development includes the interaction of three complex systems: the world economy, global society, and the geophysical environment [3]. The relationship between climate change and sustainable development has been widely discussed. For the 17 sustainable development goals committed to by the international community, there is structured evidence that climate change could undermine 16, and that addressing climate change could strengthen all 17,

but undermine efforts to achieve 12 [4]. The rapid increase in carbon emissions leading to global warming has become a focus of concern, prompting the international community to prioritize the development of a green and low-carbon economy as a consensus strategy to address and mitigate climate change [5]. Greenhouse gas emissions are primarily caused by human activities, and compared to carbon emissions from industrial production involving the burning of fossil fuels, land-use change introduces greater uncertainty into carbon emissions [6]. The Intergovernmental Panel on Climate Change (IPCC) report on climate change and land states that all scenarios that limit climate change to 1.5 °C rely heavily on methods to mitigate land-use change and decarbonize the economy [7]. Land-use change not only influences carbon emissions but also affects the stability of land ecosystems by altering their structure and function [8], which is critical for maintaining ecosystem services [9]. Maximizing environmental benefits is one of land use optimization's primary goals, and one indicator of the advantages to the environment is the value of ecosystem services [10]. The clear contrast of financial expenses and environmental advantages is another point of emphasis for economists [11]. Climate science and climate economics can help us find ways to achieve sustainable development goals [12]. For example, carbon-based social cost modeling is used to determine the best emission reduction path to cope with climate change [13]. The SCC is an estimate of the economic loss caused by emitting one additional ton of carbon dioxide [14]. Various environmental policies and green investment projects can be evaluated using the SCC. The valuation of ecosystem service values and carbon emissions provides the basis for national and governmental management decisions in the areas of human well-being and mitigating and adapting to climate change [1]. In the context of China's double carbon policy and its simultaneous emphasis on improving the ecological environment, studying the spatial patterns of carbon emissions and ecosystem service values is of great importance in addressing the challenge of balancing economic growth and environmental protection in the process of societal development.

Ecosystem services refer to the range of benefits that humans directly or indirectly derive from the environment [15]. Ecosystem service functions are divided into four categories by the Millennium Ecosystem Assessment report: providing, regulating, sustaining, and cultural services [16]. The valuation of ecosystem services provides a clear calculation of the benefits that humans derive from ecosystem services, thus supporting informed decision-making [17]. Common methods for estimating the value of ecosystem services can be categorized as energy value analysis, ecological space valuation, material quality valuation, and value quantity valuation [18]. Among them, the value quantity evaluation method, which is determined by each land use type's area and its equivalent factor value coefficient, has been widely applied due to its ease of data collection and low estimation cost [19]. The model for estimating ESV that Costanza et al. [20] proposed in 1997 has gained widespread recognition and use. Xie et al. [21] redefined the classifications of ecosystem services in light of China's ecological traits and improved the ESV equivalent factor for different land use types. Given China's vast geographical diversity and complex ecosystems, the application of ESV assessment needs to take scale into account. Studying the area's natural environment and socioeconomic circumstances necessitates creating a tailored ESV assessment system by adjusting ESV equivalent components per unit area [22].

Land application activities, along with processes that result in carbon emissions, are referred to as being transformed by the human production activities they support, which release carbon dioxide into the atmosphere, a process that includes both direct and indirect carbon emissions. Use of land, refers to the processes, actions, and tactics by which land is subject to carbon emissions [6]. The carbon balance of terrestrial ecosystems is directly influenced by changes in land use, and this has an impact on regional carbon emission levels [23], thus having a major impact on the processes involved in the global carbon cycle [24]. The land inventory approach, mechanism model simulation method, and carbon emission coefficient method are some of the techniques used to account for carbon emissions. The emission coefficient method is particularly straightforward and may be applied at many different scales [25]. In addition to offering academic references for

reducing global warming, the assessment of land use carbon emissions in the study area is useful for adaptive land use planning and management [26]. The best avenues for the regulation and optimization of land use carbon emissions [27] and control [28] have crucial practical significance.

Additionally, many investigations have been done, confirming the impacts of altering land use on carbon emissions and ecosystem service values [29–31], and current studies typically take both into account to provide evidence for sustainable environmental management. For example, Soumik Saha et al. [32] quantified carbon stocks, ecosystem service value status, and total primary productivity in the Chota Nagpur Plateau (India). Chen et al. [33] studied the spatial and temporal differences in carbon emissions and ecosystem survival value (ESV) caused by land use cover changes in the Chengdu–Chongqing urban agglomeration in China. In addition, some scholars have investigated the quantitative [34] and spatial [35] relationships between carbon emissions based on land use and the value of ecosystem services. For example, Du et al. [36] calculated and analyzed the ecological and environmental impacts of land use change in Hangzhou, including ecosystem service value (ESV) and carbon emissions. Yang et al. [37] evaluated the carbon emission intensity and ESV intensity of the Guanzhong Plain urban agglomeration in China during 2000–2020, and concluded there was a substantial negative spatial connection between the two [38]. The use of SCC to represent the correlation between carbon emissions and ESV is compelling and comparable, because both are methods of quantitative evaluation from the perspective of monetary value [39]. SCC accounting is often used as evidence for future climate policy and for optimizing the structure of carbon emissions [40]. For example, individual countries' $CO_2$ emissions from fossil fuels and industrial processes have resulted in a reduction in global wealth from 1950 to 2018, as estimated by Wilfried Rickels et al. [41] through a historical time series of the social cost of carbon. Payments for carbon and payments for ecosystem services have been evaluated to compensate for the loss of livestock income due to reduced grazing regimes, and to provide carbon sequestration and other benefits [42], but there are few studies on the relationship between carbon emissions and the value of ecosystem services in island ecosystems.

The island ecosystem is a special and complex region located at the interface between the ocean and the land. The ecological structure of the island is simple, land resources are limited, species richness is low, and the ecology is fragile [43], yet island ecosystems have significant impacts on the global carbon cycle [44]. At present, there are few studies assessing the carbon emissions [45] and ESV [46,47] of island ecosystems.

This study aims to determine the spatial and temporal evolution and distribution of ESV and SCC in island ecosystems induced by land use change, and to support the implementation of integrated policies for land use management and carbon emission control. Taking the Zhoushan Archipelago as an example, we spatially quantify ESV and SCC from 2010 to 2020, and analyze their spatial and temporal distribution characteristics and evolution patterns. Next, in order to figure out whether there is a spatial correlation relationship between ESV and SCC, we perform a spatial correlation investigation between the two via the grid, their spatial interaction characteristics, and local clustering patterns. The research conclusions have significant theoretical and practical implications for encouraging low-carbon, green, and high-quality regional development. Overall, this supports ecological security and economic decarbonization, assures human well-being, and helps the archipelago region develop in a way that is both environmentally and economically viable.

## 2. Materials and Methods

### 2.1. Study Area

South of the Yangtze River estuary and on the outskirts of Hangzhou Bay in the East China Sea lies the Zhoushan Archipelago in China's southeast. The geographical location of the study area is displayed in Figure 1. Zhoushan is the first prefecture-level city in China to be organized as an archipelago, comprising 2085 islands, of which 141 are inhabited. It consists of Dinghai and Putuo districts and Daishan and Shengsi coun-

ties. Encompassing a total land area of 1458.76 square kilometers and a maritime area of 20,800 square kilometers [48], the Zhoushan Cross-Sea Bridge was officially opened in late 2009, and the Zhoushan Archipelago New Area in Zhejiang was established in mid-2011, accelerating the process of urbanization and industrialization in the Zhoushan Archipelago. Rapid industrial and economic development, coupled with constraints such as limited land resources and natural disasters, underscores the importance of assessing and protecting the fragile ecosystems of these islands.

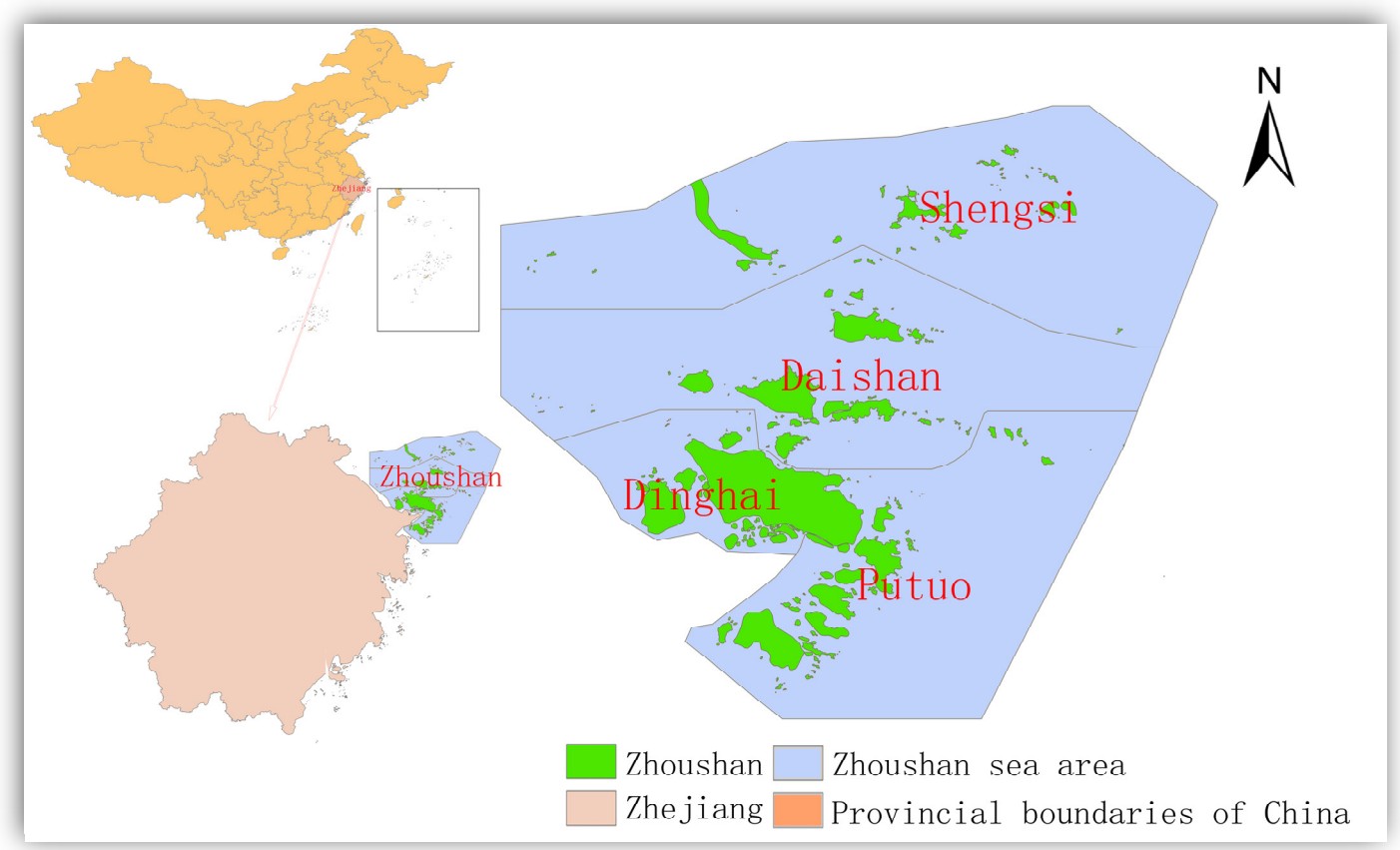

**Figure 1.** Geographical location map of the Zhoushan Archipelago.

### 2.2. Data Sources and Processing

The types of data used in this analysis were land use data, carbon emission data, and the ESV coefficient. Zhoushan City's land use and administrative division statistics were sourced from the Zhejiang Provincial Geographic Information Public Service Platform [49] and the Chinese Academy of Sciences Resource and Environmental Science and Statistics Center [50], respectively. For analysis, the land classes in the research region were reclassified into eight categories: farmland, forest, shrubland, grassland, waterbody, construction land, unused land, and mudflat. ESV coefficients, carbon emission coefficients, and correction factors were obtained from the Zhoushan Archipelago Bureau of Statistics [48], the National Bureau of Statistics [51], and relevant published literature, as described in Section 2.3.

### 2.3. Research Methods
#### 2.3.1. Carbon Emission Calculation

Among the eight land-use types in the Zhoushan Archipelago, forest, shrubland, grassland, waterbody, unused land, and mudflat serve as carbon sinks, while farmland and construction land act as carbon sources. The total areas under each type of land use and

the associated carbon emission coefficients were used to compute the carbon emissions for the Zhoushan Archipelago. The following is the calculation formula:

$$C_i = A_i \times CEC_i \tag{1}$$

In the formulas, $i$ is the type of land use, $C_i$ is the carbon emission (t/t) of class $i$ land in the Zhoushan Archipelago, $A_i$ is the land area (ha), and $CEC_i$ is the carbon emission coefficient (t/ha). The appropriate literature was consulted to derive the carbon emission coefficients for different forms of land use. According to the principle of similar latitude and longitude and similar climatic conditions, the most extensive reference values were selected to determine the carbon emission coefficient of each land-use type, with values as follows: farmland (0.422) [52], forest (−0.581) [53], shrubland (−0.161) [54], grassland (−0.021) [55], waterbody (−0.253) [52], unused land (−0.005) [56], and mudflat (−1.538) [57].

It is worth noting that the carbon emissions of construction land mainly come from the processes of human production and life. With the acceleration of the urbanization process, the carbon-emitting activities on construction land have increased notably, and the carbon emission coefficient shows significant differences in different years. Considering data availability, the carbon emissions for construction land were computed by multiplying the gross output value of the secondary and tertiary industries in the Zhoushan Archipelago by their unit of gross domestic product (GDP) energy consumption [58]. The calculation formula is as follows:

$$C_S = GDP_{2,3} \times I \times K \tag{2}$$

In the equation, $C_S$ is the land's carbon emitted from construction (t), and $GDP_{2,3}$ is the output value of the second and third industries of the Zhoushan Archipelago (CNY 10,000). This is CNY 10,000 yuan for the GDP energy usage (t standard coal per 10,000 yuan). The coal consumption carbon emission coefficient, or $K$, is fixed at 0.7476 (t/t) [59].

As of now, the average transaction price in China's eight major carbon market pilot cities is about CNY 29.37 per ton of carbon dioxide [60]. By estimating the study region's carbon emissions using Equation (3), the *SCC* of the area can be determined.

$$SCC = \sum (C_i \times 29.37) \tag{3}$$

### 2.3.2. Ecosystem Service Value Accounting

(1) ESV Accounting Coefficients

Following the studies of Xie et al. [19], the four main categories of ecosystem services identified by this study are provisioning, regulating, sustaining, and cultural services. These four core categories are then further broken down into eleven subordinate categories. Water supply, raw material production, and food production are examples of provisioning services. Gas regulation, climate regulation, environmental purification, and hydrological regulation are examples of regulating services. Soil conservation, biodiversity preservation, and nutrient cycling maintenance are examples of supporting services. Aesthetic landscape services comprise the majority of cultural services.

The Zhoushan Archipelago ESV Equivalence Factor Table is based on the research of Xie et al. [61]. In light of the real land use conditions seen in the research region and the equivalency table's comparison of landscape and ecosystem categories, the ecosystem service values for each type are estimated. Consequently, in accordance with the table of equivalent Chinese coefficients for ESV per unit area, farmland is associated with "dry land", forest with "mixed forest", shrubland with "shrubland", grassland with "meadow", waterbody with "water", unused land with "barren" and mudflat with "wetland", while construction land does not provide ecosystem service values. A table of the base equivalent coefficients of ecosystem service values per unit area is displayed in Table 1.

**Table 1.** Ecosystem service basis equivalent value per unit area.

| Ecosystem Classification | Secondary Type | Farmland | Forest | Shrubland | Grassland | Waterbody | Construction Land | Unused Land | Mudflat |
|---|---|---|---|---|---|---|---|---|---|
| Provisioning services | Food production | 0.85 | 0.31 | 0.19 | 0.38 | 0.8 | 0 | 0 | 0.51 |
| | Raw material production | 0.4 | 0.71 | 0.43 | 0.56 | 0.23 | 0 | 0 | 0.5 |
| | Water supply | 0.02 | 0.37 | 0.22 | 0.31 | 8.29 | 0 | 0 | 2.59 |
| Regulating services | Gas regulation | 0.67 | 2.35 | 1.41 | 1.97 | 0.77 | 0 | 0.02 | 1.9 |
| | Climate regulation | 0.36 | 7.03 | 4.23 | 5.21 | 2.29 | 0 | 0 | 3.6 |
| | Environmental purification | 0.1 | 1.99 | 1.28 | 1.72 | 5.55 | 0 | 0.1 | 3.6 |
| | Hydrological regulation | 0.27 | 3.51 | 3.35 | 3.82 | 102.24 | 0 | 0.03 | 24.23 |
| Supporting services | Soil conservation | 1.03 | 2.85 | 1.72 | 2.4 | 0.93 | 0 | 0.02 | 2.31 |
| | Nutrient cycling maintenance | 0.12 | 0.22 | 0.13 | 0.18 | 0.07 | 0 | 0 | 0.18 |
| | Biodiversity maintenance | 0.13 | 2.6 | 1.57 | 2.18 | 2.55 | 0 | 0.02 | 7.87 |
| Cultural services | Providing aesthetic landscape | 0.06 | 1.14 | 0.69 | 0.96 | 1.89 | 0 | 0.01 | 4.73 |

(2)    ESV Coefficient Adjustment

The reference equivalent factor is the ecosystem service value per unit area in China. In order to improve adaptability, it is necessary to convert the national-scale equivalent factor values to the regional scale when applying them to specific events. When applying these coefficients, changes are made in accordance with the real circumstances of the research area, taking into account correction factors for crop yield, socio-economic development, and resource scarcity.

The economic worth of annual cereal production per hectare in areas with the national average yield determines the corresponding factor values that are employed. These values serve as key references and form the basis for regional corrections for other types of land use. The ratio of the average cereal yield in the study area to the average cereal yield across the country serves as the basis for the cereal yield adjustment. This correction, based on the ratio of cereal yields, reflects regional differences in the total ecological service value compared to the national average [62]. The correction formula is as follows:

$$A = \frac{Q_{ZS}}{Q} \tag{4}$$

Here, $A$ is the Zhoushan Archipelago grain yield correction coefficient, and $Q_{ZS}$ and $Q$ represent the per-unit area grain yield in Zhoushan Archipelago and the national average, respectively. According to statistical yearbooks, $A$ is calculated as 0.94 for the years 2005–2020.

The calculation of the socio-economic development coefficient takes into account both willingness and financial capacity. The percentage of total personal consumption expenditure that goes toward food is represented by the Engel coefficient. A lower Engel coefficient indicates a lower cost of living, a higher willingness to consume non-food items, and a higher willingness to pay. Therefore, the Engel coefficient is used to measure the willingness to pay, and GDP per capita represents the ability to pay. The following formula is used to modify the environmental service value according to the socioeconomic development coefficient:

$$S = \frac{EL_{ZS}}{EL} \times \frac{\overline{GDP_{ZS}}}{\overline{GDP}} \tag{5}$$

Here, $S$ is the socioeconomic development correction coefficient, $EL_{ZS}$ and $EL$ represent the Engel coefficients for the Zhoushan Archipelago and the national average, respectively, and $\overline{GDP_{ZS}}$ and $\overline{GDP}$ represent the per capita GDP for the Zhoushan Archipelago

and the national average, respectively. According to statistical yearbooks, $S$ is calculated as 1.89 for the years 2005–2020.

An indicator of the link between supply and demand for ecosystem services is the resource scarcity coefficient, with higher scarcity indicating greater demand for services than supply. This coefficient is determined by measuring resource scarcity using population density. The adjustment formula is:

$$R = \frac{\ln P_{ZS}}{\ln P} \tag{6}$$

Here, $R$ is the resource scarcity correction coefficient, while $\ln P_{ZS}$ and $\ln P$ represent the population density in the Zhoushan Archipelago and the national average, respectively. According to statistical yearbooks, $R$ is calculated as 0.86 for the years 2005–2020.

The calculation formula for the Zhoushan Archipelago standard equivalent factor correction coefficient ($EF_{ZS}$) is:

$$EF_{ZS} = A \times S \times R \tag{7}$$

The harsh climate and soil conditions on islands limit the development of island vegetation, resulting in differences in vegetation biomass between island and mainland ecosystems. Further adjustments are made to the ecosystem service equivalent of the Zhoushan Archipelago's forests:

$$EF_{f\_ZS} = \frac{b}{B} \times EF_{forset} \tag{8}$$

Here, $EF_{f\_ZS}$ is the adjusted per-unit area forest ESV equivalent, $b$ is the forest biomass in the Zhoushan Archipelago, $B$ is the average biomass of forest ecosystems per unit area in China, and $EF_{forset}$ is the ecosystem service equivalent for Chinese forests.

(3)   ESV for Various Land Use Types in the Zhoushan Archipelago

Ecosystem service value equivalent (ESVE) refers to the potential capacity of ecosystems to generate their proportionate benefits, with a factor of 1 standard unit of ESVE, i.e., the economic value of the annual natural food production of 1 hectare of farmland with a national average yield of 1/7 of the market value of the average local food yield [63]. The formula is:

$$E_V = \frac{1}{7} \times P \times Q \tag{9}$$

Here, $E_V$ is the ESV for one standard equivalent factor, $P$ is the average grain price (CNY/kg), and $Q$ is the grain yield per unit area (kg/ha). According to the Zhoushan Archipelago Statistical Yearbook, the average grain yield per unit area during the study period was 4939.81 kg per ha. Additionally, the average price of grain in Zhoushan Archipelago was CNY 2.1 per kg. Based on calculations, the Zhoushan Archipelago's ecosystem service value for one standard equivalent factor is CNY 1481.94 per ha.

The basic ESV equivalent factor was corrected according to Equation (7), and then the forest ESV equivalent factor was further corrected by Equation (8). Based on the ESV of one standard equivalent factor in the Zhoushan Archipelago, the ESV per ecological unit area ($ESV_{ZS}$) was calculated, and the results are displayed in Table 2.

$$ESV_{ZS} = \sum AREA \times E_V \tag{10}$$

**Table 2.** Ecosystem service value coefficient of the Zhoushan Archipelago (unit: CNY /ha).

| Ecosystem Classification | Secondary Type | Farmland | Forest | Shrubland | Grassland | Waterbody | Construction Land | Unused Land | Mudflat |
|---|---|---|---|---|---|---|---|---|---|
| Provisioning services | Food production | 1924.59 | 540.47 | 430.2 | 860.4 | 1811.38 | 0 | 0 | 1154.75 |
| | Raw material production | 905.69 | 1237.85 | 973.61 | 1267.96 | 520.77 | 0 | 0 | 1132.11 |
| | Water supply | 45.28 | 645.08 | 498.13 | 701.91 | 18,770.39 | 0 | 0 | 5864.33 |
| Regulating services | Gas regulation | 1517.03 | 4097.11 | 3192.55 | 4460.51 | 1743.45 | 0 | 45.28 | 4302.02 |
| | Climate regulation | 815.12 | 12,256.45 | 9577.65 | 11,796.59 | 5185.07 | 0 | 0 | 8151.19 |
| | Environmental purification | 226.42 | 3469.47 | 2898.2 | 3894.46 | 12,566.42 | 0 | 226.42 | 8151.19 |
| | Hydrological regulation | 611.34 | 6119.51 | 7585.14 | 8649.32 | 23,1493.9 | 0 | 67.93 | 54,862.06 |
| Supporting services | Soil conservation | 2332.15 | 4968.83 | 3894.46 | 5434.13 | 2105.73 | 0 | 45.28 | 5230.35 |
| | Nutrient cycling maintenance | 271.71 | 383.56 | 294.35 | 407.56 | 158.5 | 0 | 0 | 407.56 |
| | Biodiversity maintenance | 294.35 | 4532.97 | 3554.83 | 4936 | 5773.76 | 0 | 45.28 | 17,819.42 |
| Cultural services | Providing aesthetic landscape | 135.85 | 1987.53 | 1562.31 | 2173.65 | 4279.38 | 0 | 22.64 | 10,709.76 |

### 2.4. Bivariate Spatial Correlation Analysis

A fishnet analysis tool was utilized to visually inspect the spatiotemporal patterns of the ESV and SCC in the Zhoushan Archipelago. The city was divided into a grid of 3 km × 3 km using kilometer grids. By calculating the ESV and SCC for each grid, the spatiotemporal variations in the ESV and SCC for the Zhoushan Archipelago were obtained. This stage was carried out using ArcGIS 10.4.

The bivariate Moran's index was then employed to investigate the spatial dispersion and clustering between ESV and SCC. The global bivariate Moran's I examined whether there was a spatial correlation, and the degree thereof, between carbon emission social cost intensity and ESV intensity across the region. Meanwhile, the local bivariate Moran's I showed spatial correlations in specific grid cells [64]. The analysis was carried out using Geoda 1.16.

### 3. Results

#### 3.1. Spatiotemporal Evolution of Land Use Structure in the Zhoushan Archipelago

As shown in Table 3, the area devoted to each type of land use was meticulously recorded. The unit is hectares.

**Table 3.** Temporal changes in land type area in Zhoushan Archipelago, 2010–2020 (unit: hectare).

| Land Use Type | Year | | |
|---|---|---|---|
| | **2010** | **2015** | **2020** |
| Farmland | 25,445.86 | 25,694.56 | 25,609.90 |
| Forest | 71,758.81 | 70,961.86 | 70,919.01 |
| Shrubland | 3660.83 | 3763.73 | 3763.66 |
| Grassland | 1270.99 | 2302.81 | 2302.81 |
| Waterbody | 6270.76 | 6342.48 | 5904.85 |
| Construction land | 25,728.66 | 28,599.03 | 29,441.51 |
| Unused land | 20.50 | 20.91 | 20.91 |
| Mudflat | 4077.15 | 4411.54 | 4133.77 |
| Total land area | 138,233.57 | 142,096.94 | 142,096.42 |

Our land use classification results show that the total land area of the Zhoushan Archipelago increased between 2010 and 2015, mainly due to human reclamation activities. The area decreased slightly from 2015 to 2020, with relatively small changes. The research area selected in this paper is the archipelago. Unlike inland cities, the total amount of land

in island areas will change due to human reclamation activities and changes in coastlines. As the local government stopped the sea from landing, the total amount of land gradually stabilized, but the overall land size in the limited area fluctuated due to the shift in coastline. Every year, the area set aside for construction increased while the area covered by water and forests gradually shrank. Influenced by human reclamation activities on mudflats, grassland and mudflat areas showed an overall increase. Other types of land use showed relatively stable changes, with small fluctuations in their contribution rates. Figure 2 clearly depicts the direction of land use transitions between different land types. Land use types changed frequently between 2010 and 2015, with shifts occurring in all types of land. Among them, the destination and source of shifts in construction land are the most important, followed by the waterbody. During 2015–2020, changes occurred only in construction land, and the sources were farmland, forests, water bodies, and mudflats.

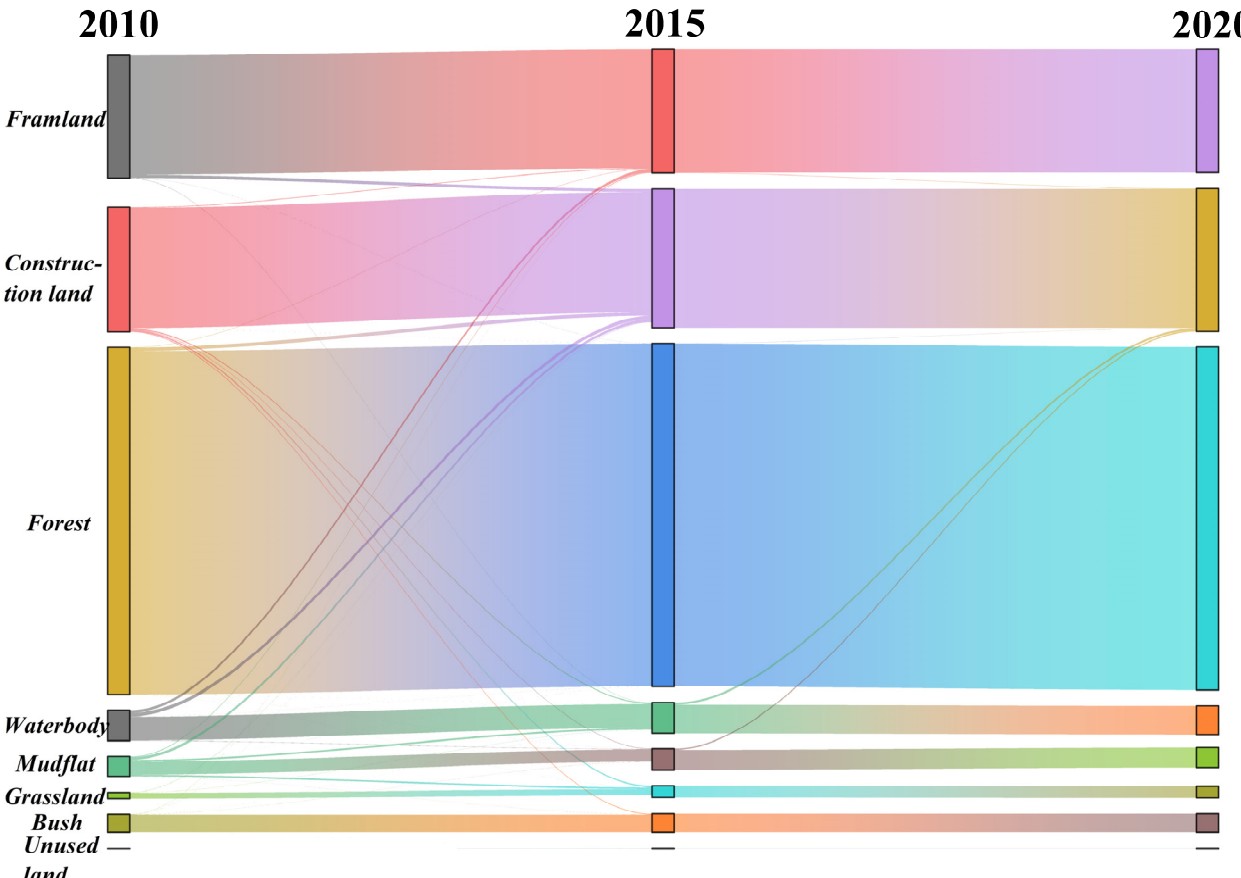

**Figure 2.** Land conversion Sankey diagrams.

*3.2. Spatiotemporal Evolution of SCC*

Utilizing the area of each land-use type and the carbon emission coefficients for the Zhoushan Archipelago, the carbon emissions for each land type during 2010–2020 were calculated using Formulas (1) and (2). The results are presented in Table 4. According to Equation (3), carbon emissions have been used to calculate the SCC (Table 5). The SCC is calculated using carbon emissions, and the value for each grid is the total value of the SCC for all land classes within that grid. It is split into five display levels, each spaced equally between low and high, and the results of its spatial and temporal changes are shown in Figure 3.

**Table 4.** Temporal changes of carbon emissions in the Zhoushan Archipelago, 2010–2020 (unit: t).

| Land Use Type | Year | | |
|---|---|---|---|
| | **2010** | **2015** | **2020** |
| Farmland | 10,738.15 | 10,843.11 | 10,807.38 |
| Forest | −41,691.87 | −41,228.84 | −41,203.95 |
| Shrubland | −589.39 | −605.96 | −605.95 |
| Grassland | −26.69 | −48.36 | −48.36 |
| Waterbody | −1586.50 | −1604.65 | −1493.93 |
| Construction land | 571,493.23 | 1,166,625.60 | 16,078,977.24 |
| Unused land | −0.10 | −0.10 | −0.10 |
| Mudflat | −6270.65 | −6784.95 | −6357.74 |
| Total | | | |
| Sink | 50,165.21 | 50,272.87 | 49,710.03 |
| Source | 582,231.38 | 1,177,468.70 | 16,089,784.62 |
| Net carbon emission | 532,066.17 | 1,127,195.83 | 16,040,074.59 |

**Table 5.** SCC for each category in the Zhoushan Archipelago, 2010–2020 (unit: CNY).

| Land Use Type | Year | | |
|---|---|---|---|
| | **2010** | **2015** | **2020** |
| Farmland | 315,379.47 | 318,462.14 | 317,412.75 |
| Forest | −1,224,490.22 | −1,210,891.03 | −1,210,160.01 |
| Shrubland | −17,310.38 | −17,797.05 | −17,796.75 |
| Grassland | −783.89 | −1420.33 | −1420.33 |
| Waterbody | −46,595.51 | −47,128.57 | −43,876.72 |
| Construction land | 16,784,756.17 | 34,263,793.87 | 472,239,561.54 |
| Unused land | −2.94 | −2.94 | −2.94 |
| Mudflat | −184,168.99 | −199,273.98 | −186,726.82 |
| Total | 15,626,783.71 | 33,105,742.11 | 471,096,990.71 |

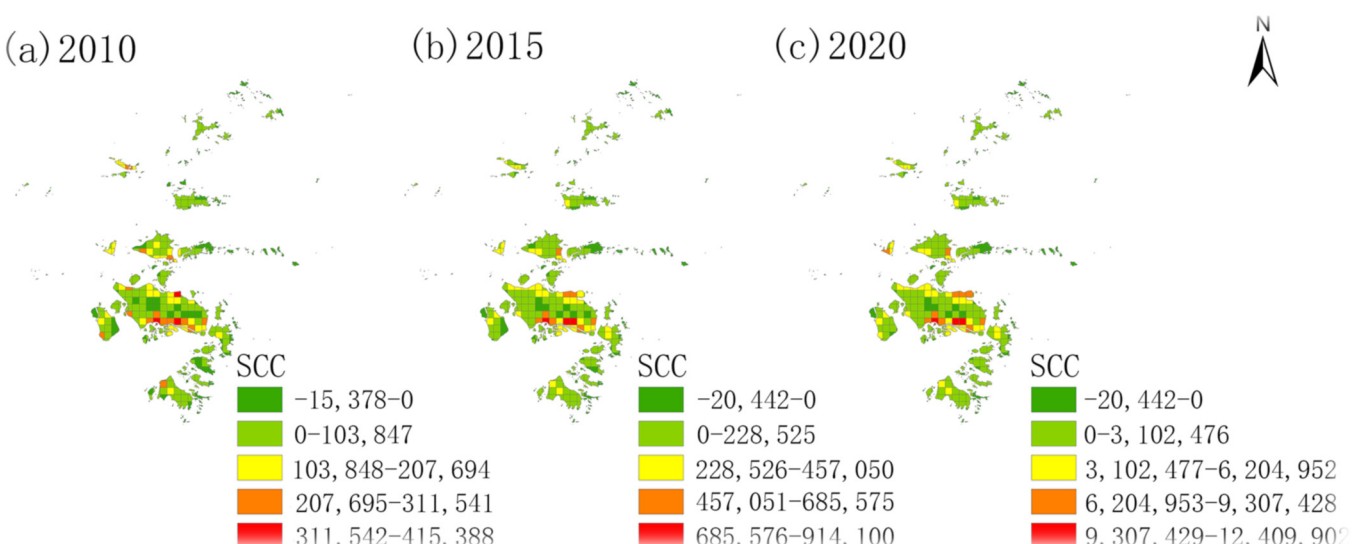

**Figure 3.** Temporal and spatial changes of SCC in Zhoushan Archipelago.

The carbon emissions results show that the net carbon emissions in the Zhoushan Archipelago were consistently positive from 2010 to 2020, reflecting an overall trend in carbon emissions. Between 2010 and 2020, the net carbon emissions showed a significant upward trend, increasing from $53.21 \times 10^4$ t to $1604.01 \times 10^4$ t. The growth trend slowed down between 2015 and 2020. The carbon source showed a similar trend to the net carbon emissions, with a slight decrease in carbon sinks. Looking at the carbon source composition,

carbon emissions from construction land increased from $1057.91 \times 10^4$ t to $4125.12 \times 10^4$ t, making this the main carbon source. The Zhoushan Archipelago saw a sharp rise in urbanization and industrialization after 2010, which was mostly responsible for the rise in carbon emissions from building sites. The building land's yearly carbon emissions and the regional energy consumption pattern, represented by the per capita carbon emissions from construction land, have shown rapid growth trends since 2015; the annual per-person carbon emissions from building sites are 22.21, 40.79, and 546.13 t/ha, respectively. Looking at the composition of carbon sinks, forests are the main carbon sink, followed by mudflats. The share of each land category remains relatively constant. Table 5 shows that the carbon sink category has a negative SCC, meaning that it equalizes the value of the damage caused by each ton of carbon dioxide emitted into the atmosphere. The SCC increased from CNY 15 million in 2010 to CNY 471 million in 2020.

As can be seen in Figure 3, the value for each grid cell represents the aggregated SCC value in CNY for all land classes within this grid range. A negative number for the first rank indicates the SCC compensation that these grid cells can provide. The low SCC areas are mainly distributed in the central area of Zhoushan Island, the coastal fringe of larger islands, and some fragmented small islands far from land, while all other classes show positive numbers. Specifically, the areas with higher SCC are mainly located in Zhoushan Island, Daishan Island, Liuhang Island, and Jintang Township, which are the four regions with large areas and the most concentrated population. Yuoshan Island, to the left of Daishan Island, is a typical industrial island, and its carbon emissions have increased steadily over the study period. The scattered islands in the periphery, far from the mainland, are low-carbon-emission areas, benefiting from their small size, resistance to development, and sparse human activities.

### 3.3. Spatiotemporal Evolution of Ecosystem Service Value

Using Formula (9), the area of each land type and its ESV were used to compute the spatiotemporal variations in ESV in the Zhoushan Archipelago between 2010 and 2020, as shown in Table 6. The ESV temporal changes categorized by land-use type are presented in Table 7. ESV has been divided into five classes, and Figure 4 displays the outcomes.

**Table 6.** Temporal changes in ESV for functional division, 2010–2020 (unit: million CNY).

| Secondary Ecosystem Classification | Year | | |
|---|---|---|---|
| | **2010** | **2015** | **2020** |
| Food supply | 106.49 | 107.99 | 106.69 |
| Raw material | 124.93 | 125.99 | 125.32 |
| Water supply | 191.77 | 195.35 | 185.48 |
| Gas regulation | 378.44 | 382.04 | 379.78 |
| Climate regulation | 1016.05 | 1022.74 | 1017.62 |
| Purifying environment | 382.33 | 387.56 | 379.63 |
| Hydrology adjustment | 2168.77 | 2208.70 | 2091.84 |
| Soil conservation | 471.60 | 476.12 | 473.34 |
| Nutrient cycling | 38.69 | 39.05 | 38.83 |
| Biodiversity protection | 460.92 | 469.21 | 461.51 |
| Cultural and amenity services | 225.06 | 229.80 | 224.86 |
| Total | 5565.04 | 5644.56 | 5484.88 |

According to the data, the Zhoushan Archipelago's total ESV fluctuated between 2010 and 2020, characterized by an initial increase followed by a subsequent decrease. Overall, there was a decreasing trend, from CNY 5565.04 million in 2010 to CNY 5484.88 million in 2020, representing a decrease of 1.5%. In terms of individual ESV functionalities, hydrological regulation and climate regulation emerged as major contributors, together accounting for 57% of the total. Climate regulation increased, hydrological regulation decreased, and other functions remained stable. Looking at the different land use types, woodland and

watershed are the main contributors to ESV, both accounting for 83% of the total, followed closely by mudflat. Grassland increased, watersheds decreased, and other types fluctuated within a 1% range.

**Table 7.** Temporal changes in ESV by land use type division, 2010–2020 (unit: million CNY).

| Land Use Type | Year | | |
|---|---|---|---|
| | **2010** | **2015** | **2020** |
| Farmland | 231.04 | 233.29 | 232.53 |
| Forest | 2887.49 | 2855.42 | 2853.70 |
| Shrubland | 126.16 | 129.70 | 129.70 |
| Grassland | 56.66 | 102.67 | 102.67 |
| Waterbody | 1783.46 | 1803.86 | 1679.39 |
| Construction land | 0.00 | 0.00 | 0.00 |
| Unused land | 0.01 | 0.01 | 0.01 |
| Mudflat | 480.23 | 519.61 | 486.90 |
| Total | 5565.04 | 5644.56 | 5484.88 |

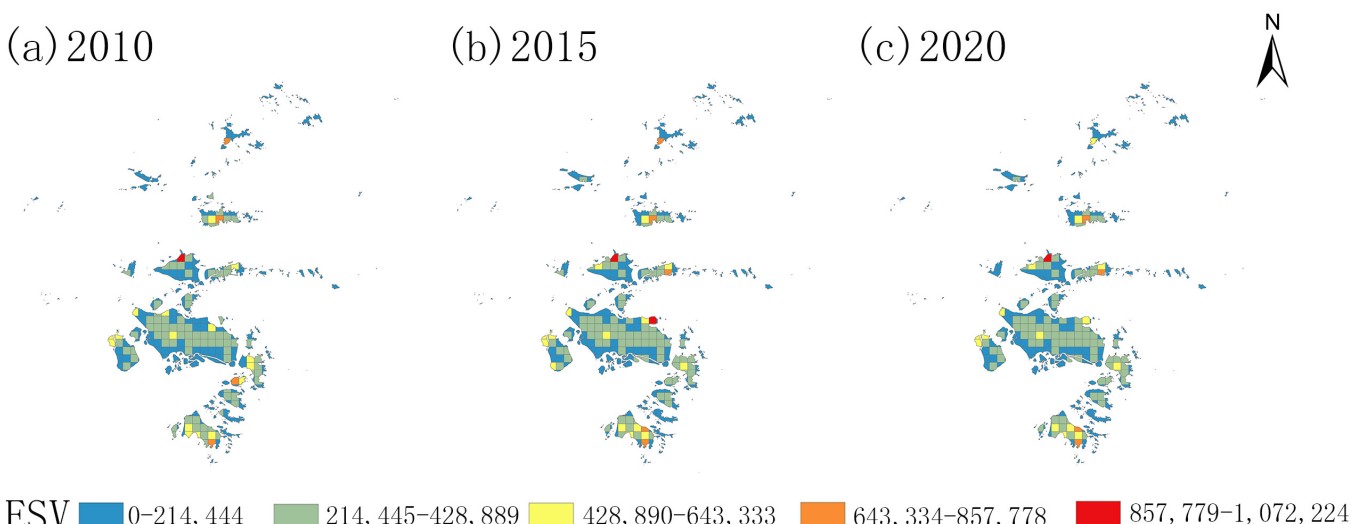

**Figure 4.** Temporal and spatial variation in ESV in the Zhoushan Archipelago.

From Figure 4, the value for each grid cell represents the aggregated ESV value in CNY for all land classes within this grid range. The Zhoushan Archipelago generally exhibits low ESV, with fewer areas of high ESV. Over the course of the study period, there was a minor change in the geographical and temporal variability of ESV in the Zhoushan Archipelago, with an overall decreasing trend. The regions with low ESV values are mainly distributed in coastal areas near large islands, which also feature aggregations of built-up areas, while there is a concentration of high ESV-regions in places with extensive forests, water, and mudflat areas, in line with the distribution of different land use types and their respective ESV contribution rates.

### 3.4. Spatial Correlation Analysis between ESV and SCC

A bivariate global Moran's I index study for the ESV and SCC of carbon emissions in the Zhoushan Archipelago from 2020 to 2020 was carried out using Geoda1.16. The analysis included 999 randomizations for testing, and Table 8 presents the findings. The Moran scatterplot is shown in Figure 5, wherein each quadrant represents High–High (H–H), Low–High (L–H), Low–Low (L–L) and High–Low (H–L) clustering types. In addition, the two-variable local spatial clustering relationship for the social cost intensity and ESV intensity of carbon emissions for each grid cell was investigated using Moran's I analysis, as shown in Figure 6.

**Table 8.** Bivariate global Moran's I index of the Zhoushan Archipelago from 2010 to 2020.

| Year | Moran's I | $p$-Value | z-Value |
|---|---|---|---|
| 2010 | 0.128 | 0.001 | 13.3427 |
| 2015 | 0.141 | 0.001 | 14.7151 |
| 2020 | 0.14 | 0.001 | 14.3616 |

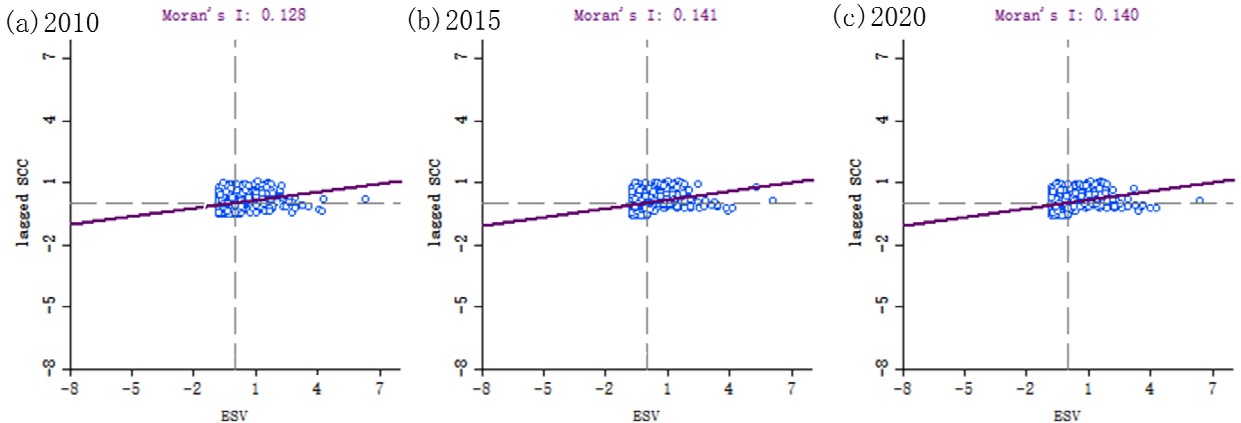

**Figure 5.** Moran scatter plot 2010–2020.

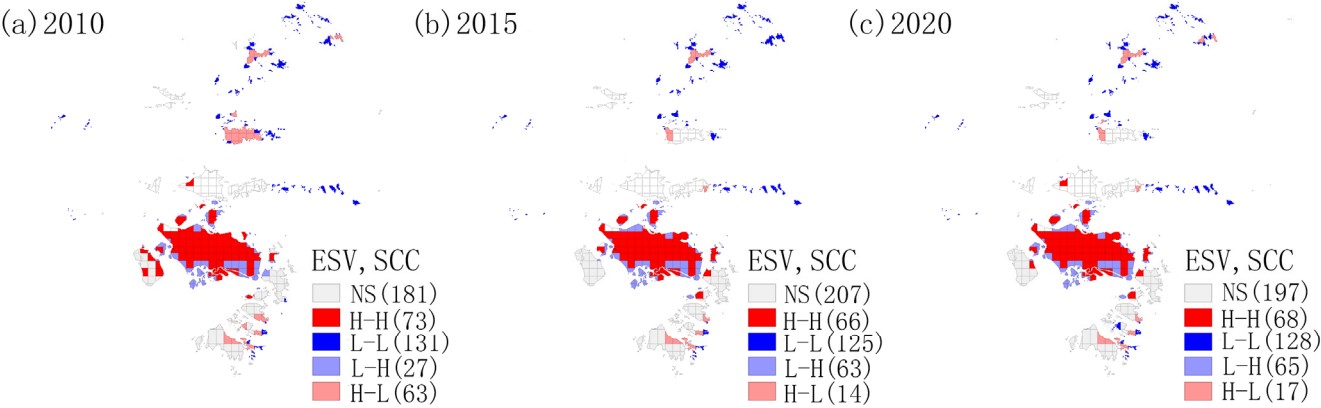

**Figure 6.** Spatial correlation between ESV and SCC.

The results show that the correlation coefficients between ESV and SCC in the Zhoushan Archipelago from 2010 to 2020 were 0.142, 0.136, and 0.134, respectively. The Moran's I value was positive, with $p$-values < 0.01 and z-values > 2.58, indicating a significant positive spatial correlation between ESV and SCC at a 99% confidence level. In other words, regions with high ESV tend to have high SCC. This does not seem to fit with our current knowledge. In general, a high ESV corresponds to a low SCC. We have explored the causes of this result through bivariate local analysis. Looking at Figure 5, ESV and SCC are clustered in the first and third quadrants, while they are dispersed in the second and fourth quadrants. This confirms that the H–H and L–L types show clustering, while the L–H and H–L types are spatially dispersed.

Figure 6 shows that the changes in clustering types in the Zhoushan Archipelago over time are minimal. Throughout the investigation, there was a decrease in the number of grids in H–H and L–L, but an increase in the quantity of grids in H–L and L–H. The regions where ESV and SCC had a positive association were H–H and L–L, whereas the regions where there was a negative correlation were L–H and H–L. The proportion of networks with a positive correlation was 24% higher than the proportion of networks with a negative correlation. H–H was mainly concentrated in Zhoushan Island, Putuoshan Island, Daishan

Island, and other economically active regions with significant carbon emissions. These areas also prioritize environmental protection, resulting in higher ESV. Some islands, such as Shengsi Archipelago and Dongji Archipelago, have L–L characteristics due to their distance from the mainland, limited human activities, and lower carbon emissions. At the same time, these areas have single vegetation types and poorer ecological environments, which contribute less to ESV. L–H was seen to be distributed in areas where construction land is accumulating on Zhoushan Island. Urbanization affects ecological resources in these regions, causing some degree of environmental degradation, resulting in lower ESV and higher carbon emissions. H–L was mainly found on Qushan Island, some islands in Shengsi County, and some islands in the eastern part of Putuo District. Local correlations in other regions were not significant (NS).

## 4. Discussion

The examination of land use changes across the research period has revealed that changes in total land area were mainly driven by the reclamation and development of the Wadden Sea. Overall, the growth of land resources on the islands is limited. While other land types, like farming, have dropped proportionately, construction land has continued to rise. The inconvenience of transport has been a major factor limiting island development. However, after 2010, the construction of cross-sea bridges in Zhoushan accelerated the development of islands. Rapid economic development and accelerated urbanization and industrialization have led to an increase in human activities in built-up areas, thereby causing a rise in carbon emissions. During the 2010–2020 study period, Zhoushan's net carbon emissions increased by $1550.8 \times 10^4$ t, and the SCC increased by 2452%. The total ESV of the Zhoushan Archipelago showed a downward trend throughout the time spent studying, with a small change of 1.5%. This is attributed to the high ESV of land types such as forests and waterbodies, which have strong ecological compensation capabilities. Mudflats, as a characteristic land type for islands, should also not be overlooked. Therefore, future planning should prioritize the preservation of forests, waterbodies, and mudflats in order to increase regional ESV and promote high-quality development.

This study was motivated by two main factors. First, previous studies have mostly treated carbon emissions and ecosystem service values as separate evaluation units to assess environmental benefits, with limited studies on their correlation [33,34,36,65]. Second, scholars have mainly focused their research on the provincial level or above, with less attention to coastal and island areas [66]. In the paper by Wang et al. [64], the analysis of carbon emissions and ESV in the Nansi Lake Basin revealed a clear negative spatial connection between the two variables' intensities, as well as evident local aggregation. Implementing strategies to slow down the increase in carbon emissions can help establish a healthy ecological cycle and pave the way for the basin's low-carbon economy to become a reality. In contrast, this study focuses on island cities, examines two separate indicators used to assess environmental benefits, ESV and SCC, and concludes that there is a strong positive spatial correlation at the 99% confidence level between the Zhoushan Archipelago's ESV and its SCC. It also passed the *p*-value test. This means that high ESV is accompanied by high SCC and low ESV is accompanied by low SCC in the Zhoushan Archipelago. However, this is contrary to the results of previous studies. To determine the causes of the variations, we examined the local spatial connection between ESV and SCC. Firstly, ESV and SCC show negative correlation in areas with high human activities, while the SCC of ESV in areas with low human footprints reflects a positive correlation, and the number of grids reflecting positive correlation in Zhoushan Archipelago is much larger than that of those with negative correlation (24% higher). This is a strong reason. Environmental constraints exist; the population density of the islands is much lower than that of inland cities; the limited land resources of the islands make it impossible to expand construction land without limitations; and the proportion of construction land is small compared with that of natural land, such as forest land. In terms of policy, the islands are short on fresh water resources and are ecologically fragile, so the government attaches importance to the

preservation of high-ESV land such as forests, waterbodies, and mudflats. Assuredly, this is evidence that the global spatial correlation between ESV intensity and SCC intensity in the Zhoushan Archipelago is idiosyncratic. The land dispersion of the archipelago region, the low adjacency between grid cells, and the long distances have an impact on the results, resulting in non-significant correlations in many areas.

Our study provides a case analysis of the spatial distribution characteristics of the social costs of land carbon emissions and ecosystem service values associated with land use change during rapid urbanization. However, there are some potential limitations that require further refinement. First, whether or not the land use types' carbon emission coefficients as derived from the literature accurately reflect the unique topography of the Zhoushan Archipelago remains an issue. Based on extensive studies that define farmland as a carbon source and refer to carbon emission coefficients, farmland could also act as a carbon sink [67]. Owing to data collection challenges, the Zhoushan Archipelago's construction sites' carbon emissions are estimated using production figures from tertiary and secondary businesses; hence, the precision of carbon accounting may not match actual conditions. Therefore, in the future, carbon emission coefficients should be determined scientifically and reasonably based on the actual situation of the region. Other land classes refer to averages for areas of similar longitude and climate to China, which reduces inaccuracy somewhat, but not quite to the level of measured data precision. Future studies should aim to assess the carbon emissions of a proprietary territory more objectively by combining data from field surveys or by utilizing new technologies like remote sensing. Another drawback of this paper is the absence of scenario analysis able to explain and defend our results in light of the planning policies in place.

## 5. Conclusions

Land use change analysis: From 2010 to 2015, land area increased due to tidal flat reclamation and land reclamation activities. Out of all the property kinds, development land showed the biggest increase. However, strict policies controlling land reclamation and limited land resources on islands prevent the unrestricted extension of building sites. The total area of the islands started to stabilize after 2015. Based on the direction of land use conversion, land use types changed frequently between 2010 and 2015, with shifts occurring in all types of land. Among them, the destination and source of shifts in construction land are the most important, followed by those for waterbodies. During 2015–2020, changes occurred only in construction land, and the sources were farmland, forests, waterbodies, and mudflats.

Spatiotemporal evolution of SCC in the Zhoushan Archipelago (a socioecological analysis). This study investigated the spatiotemporal dynamics of carbon emissions in the Zhoushan Archipelago from 2010 to 2020, and examined their associated societal costs. During this period, the net carbon emission was positive, indicating that the Zhoushan Archipelago as a whole shows carbon emissions, and the carbon emissions show a changing trend of increasing year by year. They increased from $53.41 \times 10^4$ t to $1604.01 \times 10^4$ t. Forests emerged as the primary carbon sink, followed by mudflats and waterbodies, while construction land was the largest contributor to carbon emissions. The period from 2010 to 2015 witnessed accelerated industrialization, propelling rapid growth in the secondary and tertiary sectors, resulting in a noticeable surge in carbon emissions. Spatially, carbon sink regions were concentrated in the central area of Zhoushan Island, along the coastlines of larger islands, and on smaller islands situated farther away from the mainland. The SCC in the Zhoushan Archipelago increased by 2452%, with a substantial societal impact. From a sociological perspective, areas with larger islands and abundant land and biological resources, accompanied by a higher population density, incurred elevated societal costs in terms of carbon emissions. Spatially, a limited number of grid cells exhibited negative values in SCC, suggesting the potential for SCC compensation in these areas. However, a temporal analysis revealed fewer regions with high SCC and a majority with low SCC in the Zhoushan Archipelago. The spatiotemporal evolution analysis of ESV during the

study period indicated an overall declining trend, primarily attributed to increased land development, albeit with a minimal decrease of 1.5%. The Zhoushan Archipelago generally demonstrated low ESV, with fewer regions characterized by high ESV. The distribution of ESV was closely related to land-use types, with dense urban areas exhibiting low ESV and areas designated for forestry, waterbodies, and mudflats showcasing high ESV.

Global spatial correlation analysis between ESV and SCC during the study period revealed a significant positive correlation in the Zhoushan Municipality, supported by *p*-value tests. This suggests that, contrary to common perceptions, higher ESV accompanies higher SCC, while lower ESV is associated with lower SCC. However, local spatial correlation analysis indicated distinctions between island regions and inland cities. Regions with frequent human activities displayed a negative correlation between ESV and SCC, whereas areas with minimal human impact showed a positive correlation. The number of grids exhibiting a positive correlation in the Zhoushan Archipelago far exceeded those with a negative correlation, with the contribution rate of positive correlation regions being 24% higher. Land with carbon sinks, such as forests, demonstrated higher preservation rates, with a higher proportion of land use compared to carbon source areas, such as construction land. These findings provide evidence of the unique global spatial correlation between ESV and SCC in the Zhoushan Archipelago. The study suggests new possibilities for addressing the environmental and economic challenges posed by climate change, emphasizing the need to optimize the ecological security landscape, alleviate regional carbon emission pressures, leverage regional advantages, protect forests, and develop blue carbon resources like mudflats. Collaborative efforts with neighboring urban clusters are recommended as a proactive measure. The research results hold significant theoretical and practical implications for promoting sustainable development planning in the Zhoushan Archipelago.

**Author Contributions:** Conceptualization, Q.Z. and F.G.; Methodology, Q.Z.; Software, Q.Z.; Validation, F.G. and S.Z.; Resources, B.Z.; Data curation, J.L.; Writing—original draft, Q.Z.; Writing—review & editing, H.C. and S.Z.; Visualization, Q.Z.; Supervision, K.X.; Project administration, K.X.; Funding acquisition, S.Z. All authors have read and agreed to the published version of the manuscript.

**Funding:** This study is sponsored by the National Key R&D Program of China (No. 2019YFD0901204); the Fundamental Research Funds for Zhejiang Provincial Universities and Research Institutes (No. 2021JD006); and the Key R&D Program of Zhejiang Province (No. 2019C02056).

**Institutional Review Board Statement:** Not applicable.

**Informed Consent Statement:** Not applicable.

**Data Availability Statement:** Data are contained within the article.

**Conflicts of Interest:** Author Benxuan Zhao was employed by the company Zhejiang Zhonglan Environmental Technology Co., Ltd. The remaining authors declare that the research was conducted in the absence of any commercial or financial relationships that could be construed as a potential conflict of interest.

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
