# Peer review of "Examining the Social Costs of Carbon Emissions and the Ecosystem Service Value in Island Ecosystems: An Analysis of the Zhoushan Archipelago"

_sustainability, doi:10.3390/su16020932_

Round 1
Reviewer 1 Report
Comments and Suggestions for Authors
The four ESVs are broadband terms but generally fall into determinate categories. Of the four, cultural services tend to be more difficult to define. Please be more specific here than"providing aesthetic landscape services." This could include any developed landscape such as a park, zoo, residential development, or institutional campus.
All figures - Legend text is too small to be legible.
This is a strong study overall, but the specificity of the island condition needs to be underscored as part of the contradiction in the findings. For instance, the authors used the carbon emission coefficients for various land-use types obtained from relevant published literature; however, it is not clear if any adjustment needs to be made to account for island lands, which are more limited due to geological and physical conditions.
Comments on the Quality of English LanguageMinor English improvement is needed throughout. Please be sure to give a full definition of the acronym before using letters only.
Author Response
Dear Reviewer:
We sincerely thank the editors and all reviewers for their valuable feedback on our manuscript, "Examining the Social Costs of Carbon Emissions and the Ecosystem Service Value in Island Ecosystems: An Analysis of the Zhoushan Archipelago" (ID: sustainability-2790824). They are very helpful for paper revision. Based on your suggestions, we have made extensive corrections to the previous manuscript, as follows:
[Question 1] Give a more specific definition of "aesthetic landscape services" in the context of cultural services.
Response: Thank you for kindly reminding us. We consider this to be a topic worthy of exploration.
Cultural services represent one of the four major ecosystem services proposed by multilateral environmental agreements. The United Nations' "Millennium Ecosystem Assessment Report" categorizes cultural services into heritage value, spiritual and religious services, inspiration and creativity, landscape aesthetics, and recreational and tourism values [1]. Current research has predominantly focused on recreational and aesthetic values [2]. Urban areas, protected zones, as well as the systems you mentioned, such as parks, zoos, residential development, and institutional campuses, all constitute a research framework. Protected areas and parks, in particular, demand a more nuanced analysis, as seen in studies like the economic evaluation of cultural services in the Todgha Oasis in Morocco [3]. The manuscript's focus on the Zhoushan Archipelago presents a comprehensive urban system, representing a larger framework. The ecological services value system applied is based on the model proposed by Chinese experts, including Xie et al. [4,5], tailored for China's ecosystem services. This model limits cultural services to "providing aesthetic landscape services." This framework has garnered numerous followers, myself included. Chinese scholars have paid attention to the study of cultural ecosystem services at a later time, with fewer targeted studies. Research proposal that future research in China should take cultural ecosystem services into account to build a more integrating prospect of ecosystems, from Jiang,W [6]. Our future work will focus on cultural ecosystem services to make the value of ecosystem services better. Once again, we appreciate your insights, and your input will undoubtedly contribute to further refining our work.
The references are listed at the end.
[Question 2] All figures - Legend text is too small to be legible.
Response: Thank you for your careful inspection. We updated all the figures in the manuscript and enlarged the size of the legend text. We have updated it in the revised manuscript.
[Question 3]: Whether the carbon emission coefficient of island land needs to be adjusted.
Response: Response: We express our sincere gratitude for your valuable feedback. We acknowledge that carbon accounting methods include the sampling land inventory method, the mechanism model simulation method, and the carbon emission coefficient method. The sampling land inventory method necessitates extensive on-site investigations in the research area over an extended period, while the mechanism model simulation method involves the use of technologies such as remote sensing for area detection. These two methods, though more targeted and reflective of the unique characteristics of islands, demand substantial fieldwork and data collection. This is the limit of our work. In comparison, the carbon emission coefficient method is characterized by a simpler calculation principle and enjoys the broadest application scale, hence being the chosen methodology in this manuscript. In China, numerous scholars have explored carbon emission coefficients for various land use types, yet research on the carbon emission capabilities of different land use types in island regions remains limited. Combining the practical characteristics and existing research findings and guided by the principle of selecting the most widely applicable reference values based on proximity in latitude and similarity in climatic conditions, we determined the carbon emission coefficients for various land use types. Similarly, many scholars use this principle to determine the carbon emission coefficient of the study area. Here's a typical example. In the article published in Economic Geography ( an authoritative journal from China with a composite impact factor of 8.606), the average value of the carbon emission coefficient of each land use type obtained from the published literature was calculated to calculate the land carbon emissions of each province in China [7]. We believe that this approach holds practical significance. In response to your suggestions, we have supplemented the explanations in the manuscript accordingly.
The references are listed at the end.
[Question 4]: We need to make minor improvements to the English throughout.
Response: We tried our best to improve the manuscript and made some changes to it. These changes will not affect the content or framework of the paper. We do not list the details of all the changes here, but they are highlighted in red in the revised manuscript.
[Question 5]: Before using letters, there is no complete definition of acronyms.
Response: Thank you for your careful inspection. I corrected and checked all such errors in the manuscript. Here is an example.
Line 58,page 2(Revised manuscript)
“IPCC”-> “Intergovernmental Panel on Climate Change (IPCC)”
We tried our best to improve the manuscript and made some changes to the manuscript. We appreciate your earnest work and hope that the correction will meet with approval. Once again thank you very much for your comments and suggestions.
Best wishes!
Yours sincerely,
Qian Zhou
January 14, 2024
Reference:
- Reid, W.; Mooney, H.; Cropper, A.; Capistrano, D.; Carpenter, S.; Chopra, K. Millennium Ecosystem Assessment. Ecosystems and Human Well-Being: Synthesis; 2005;
- La Rosa, D.; Spyra, M.; Inostroza, L. Indicators of Cultural Ecosystem Services for Urban Planning: A Review. Ecological Indicators 2016, 61, 74–89, doi:10.1016/j.ecolind.2015.04.028.
- Ahrabous, M.; Allali, K.; Fadlaoui, A.; Arib, F.; Dolores de-Miguel, M.; Alcon, F. Economic Valuation of Cultural Services at the Todgha Oasis, Morocco. Journal for Nature Conservation 2023, 73, 126371, doi:10.1016/j.jnc.2023.126371.
- Xie, Gao Di; Zhen, Lin; Lu, Chunxia; Xiao, Yu; Chen, Cao An expert knowledge-based approach to ecosystem service valorisation. Journal of Natural Resources 2008, 911-919.
- Xie, Gao Di; Zhang, Caixia; Zhang, Changshun; Xiao, Yu; Lu, Chunxia The value of ecosystem services in China. Resource Science 2015, 37, 1740-1746.
- Jiang, W. Ecosystem Services Research in China: A Critical Review. Ecosystem Services 2017, 26, 10–16, doi:10.1016/j.ecoser.2017.05.012.
- Ji, Xueqiang; Liu, Huimin; Zhang, Yuesong Evolution of the spatial correlation network structure of carbon emissions from interprovincial land use and driving factors in China. Economic Geography 2023, 43, 190-200, doi:10.15957/j.cnki.jjdl.2023.02.020.
Reviewer 2 Report
Comments and Suggestions for Authors
This is an interesting study covering issues of influence changes in carbon emissions on ecosystem development. It was carried out in strict consistency with the stated methodology. However, there are several recommendations that could improve the manuscript.
- The abstract needs to be supplemented. It is necessary to show the results obtained and their significance, as well as implications and applicability. These aspects should also be shown more clearly in the manuscript itself (in the Conclusion).
The results of the Ecosystem Service Value (ESV) and Social Cost of Carbon (SCC) assessments should be more explicitly reflected in the Abstract, study findings and Conclusion.
- The manuscript provides little context for sustainability. The title and subject of the study imply that this is included in the field, but at the same time in the text of the manuscript we do not find conclusions that lead exactly to “sustainability”. This is also important in terms of the scope of the journal.
- Checking the text for plagiarism shows the presence of a high proportion of borrowings, but these borrowings are not confirmed. They are represented by common phrases and text expressions. At the same time, authors should check the text again, and if necessary, indicate the sources of text fragments. File attached.
- The manuscript may be supplemented with research limitations and assumptions. This part can be presented in Methods/or Discussion. Also, in the Discussion, the authors can evaluate how their results compare with the results of other studies, whether there are discrepancies, and what the reasons for the gap may be.

Author Response
Dear Reviewer:
We sincerely thank the editors and all reviewers for their valuable feedback on our manuscript, " Examining the Social Costs of Carbon Emissions and the Ecosystem Service Value in Island Ecosystems: An Analysis of the Zhoushan Archipelago " (ID: sustainability-2790824). They are very helpful for paper revision. Based on your suggestions, we have made extensive corrections to the previous manuscript, as follows:
[Question 1]: The abstract needs to be supplemented. It is necessary to show the results obtained and their significance, as well as their implications and applicability. These aspects should also be shown more clearly in the manuscript itself (in the conclusion). The results of the Ecosystem Service Value (ESV) and Social Cost of Carbon (SCC) assessments should be more explicitly reflected in the abstract, study findings, and conclusion.
Response: Thanks for your professional suggestions. As you suggested, we have added the significance and applicability of the results in the abstract and in the manuscript itself (in the Conclusion). We do not provide a comprehensive list of all the changes, but the revised manuscript highlights them in red.
[Question 2]: The manuscript provides little context for sustainability. The title and subject of the study imply that this is included in the field, but at the same time in the text of the manuscript we do not find conclusions that lead exactly to “sustainability”. This is also important in terms of the scope of the journal.
Response: Thank you for kindly reminding us. We have added the context of sustainability in the introduction section and the implications of our research for regional ecological and economic sustainability in the conclusion. We do not provide a comprehensive list of all the changes, but the revised manuscript highlights them in red. Once again, we appreciate your insights, and your input will undoubtedly contribute to further refining our work.
[Question 3]: Checking the text for plagiarism shows the presence of a high proportion of borrowings, but these borrowings are not confirmed. They are represented by common phrases and text expressions. At the same time, authors should check the text again, and if necessary, indicate the sources of text fragments. File attached.
Response: Thanks for your careful checks. The phrases and sentences with a high borrowing ratio have been optimized. We do not provide a comprehensive list of all the changes, but the revised manuscript highlights them in red.
[Question 4]: The manuscript may be supplemented with research limitations and assumptions. This part can be presented in Methods/or Discussion. Also, in the Discussion, the authors can evaluate how their results compare with the results of other studies, whether there are discrepancies, and what the reasons for the gap may be.
Response: Thank you for your constructive feedback, it was great to do and gave us new ideas. We added research limitations to the discussion. The discussion section also adds the results of comparison with other scholars' research results and analysis. The hypothesis can be realized through scenario simulation analysis, and we need to verify the data before we can supplement and improve it. The current time is limited, and we will conduct in-depth discussion based on this opinion in our future work.
[Question 5]: We need to make minor improvements to the English throughout.
Response: We tried our best to improve the manuscript and made some changes to it. These changes will not affect the content or framework of the paper. We do not list the details of all the changes here, but they are highlighted in red in the revised manuscript.
We tried our best to improve the manuscript and made some changes to the manuscript. We appreciate your earnest work and hope that the correction will meet with approval. Once again thank you very much for your comments and suggestions.
Best wishes!
Yours sincerely,
Qian Zhou
January 14, 2024
Reviewer 3 Report
Comments and Suggestions for Authors
Studying the correlation between the ecosystem service value (ESV) and the social cost of carbon (SCC) can provide important theoretical and practical significance for promoting regional low-carbon and green development. The paper took Zhoushan Archipelago as a case study, considered the ESV and SCC in Zhoushan Archipelago during the period 2010-2020, analysed their spatial development characteristics, and analysed the correlation between the two in time and space. The article has clear ideas, appropriate methods, and reliable conclusions. However I think there are one areas that needs to be supplemented.
1. The ESC and SCC are estimated based on land use. With social development the land use changed significantly thus caused the change of the ESC and SCC. However, there is only the results of land use in Several years, lack of mutual transfer of land use. I suggest adding a land use transfer matrix which enable readers to have a better understanding of the reasons for the changes in ESC and SCC.
Comments on the Quality of English LanguageMinor editing of English language required
Author Response
Dear Reviewer:
We sincerely thank the editors and all reviewers for their valuable feedback on our manuscript, " Examining the Social Costs of Carbon Emissions and the Ecosystem Service Value in Island Ecosystems: An Analysis of the Zhoushan Archipelago " (ID: sustainability-2790824). They are very helpful for paper revision. Based on your suggestions, we have made extensive corrections to the previous manuscript, as follows:
[Question 1]: lack of mutual transfer of land use.
Response: We express our sincere gratitude for your valuable feedback, which has enhanced the coherence and rigor of our manuscript. We wholeheartedly agree with your perspectives. In order to present a clear visualization of where the land use conversions went during the study period, a Sankey diagram of land use conversions based on the land use conversion matrix was developed. We added interpretations and conclusions of the Sankey diagram to the manuscript. Thank you for the helpful comments. We have updated it in the revised manuscript, section 3.1.
[Question 2]: Minor editing of the English language is required.
Response: Thanks for your professional suggestions. We tried our best to improve the manuscript and made some changes to it. These changes will not affect the content or framework of the paper. We do not list the details of all the changes here, but they are highlighted in red in the revised manuscript.
We tried our best to improve the manuscript and made some changes to the manuscript. We appreciate your earnest work and hope that the correction will meet with approval. Once again thank you very much for your comments and suggestions.
Best wishes!
Yours sincerely,
Qian Zhou
January 14, 2024
Reviewer 4 Report
Comments and Suggestions for Authors
A few corrections and adding recommendations would make your research very helpful to the research community

a few corrections srequired
Author Response
Dear Reviewer:
We sincerely thank the editors and all reviewers for their valuable feedback on our manuscript, " Examining the Social Costs of Carbon Emissions and the Ecosystem Service Value in Island Ecosystems: An Analysis of the Zhoushan Archipelago " (ID: sustainability-2790824). They are very helpful for paper revision. Based on your suggestions, we have made extensive corrections to the previous manuscript, as follows:
[Question 1]: The author has not added any information.
Response: We were really sorry for our careless mistakes. Thank you for your reminder. We have added author information to the revised manuscripts.
Detail:
Line 5-14,page 1(Revised manuscript).
Qian Zhou 1, Feng Gui 1, Benxuan Zhao2, Jingyi Liu3, Huiwen Cai1, Kaida Xu4 and Sheng Zhao 1, *
1 Marine Science and Technology College, Zhejiang Ocean University, Zhoushan 316022, China; zhouqian@zjou.edu.cn (Q.Z.); fgui@zjou.edu.cn (F.G.); caihuiwen@zjou.edu.cn (H.C.)
2 Zhejiang Zhonglan Environmental Technology Co. Ltd; Wenzhou, 325000, China; zbx515@163.com
3 National Engineering Research Center for Marine Aquaculture, Zhejiang Ocean University, Zhoushan 316022, China; liujingyi@zjou.edu.cn
4 Zhejiang Province Key Laboratory of Mariculture and Enhancement, Zhejiang Marine Fisheries Research Institute, Zhoushan 316022, China; h01011@zjou.edu.cn
* Correspondence: zhaosh@zjou.edu.cn; Tel.: 13454069742
[Question 2]: Grammar and sentence expression optimization problems.
Response: We feel sorry for our carelessness. In our resubmitted manuscript, grammatical errors have been corrected and the expression of some sentences has been optimized. Thanks for this careful correction. Specific details are as follows:
Detail:
Line 41,page 1(Revised manuscript).
“share many of the same drivers.”-> “, with many of the same drivers.”
Line 58,page 2(Revised manuscript).
“IPCC”-> “Intergovernmental Panel on Climate Change (IPCC)”
Line 60,page 2(Revised manuscript).
“affects”-> “influences”
Line 62-63,page 2(Revised manuscript).
“One of the main objectives in the land-use optimisation problem is to maximise environmental benefits,”-> “Maximizing environmental benefits is one of land use optimization's primary goals,”
Line 435,page 14(Revised manuscript).
“scc”-> “SCC”
[Question 3]: To go further and state the reason.
Response: Thanks for your insightful comment. As you suggested, we have added more references to support this idea.
Detail:
Line 64-68, page 2(Revised manuscript).
Economists also emphasize the explicit comparison of economic costs and environmental benefits [11]. Climate science and climate economics can help us find ways to achieve sustainable development goals [12]. For example, carbon-based social cost modeling is used to determine the best emission reduction path to cope with climate change [13].
The references are listed at the end. Codes are executed in the order in which they appear in the revised manuscripts.
[Question 4]: Lack of reference.
Response: We feel sorry for our carelessness. We have completed the references and updated the bibliography in our resubmitted manuscript. Thanks for your correction.
Detail:
Line 95-97,page 2-3(Revised manuscript).
Land application activities, along with processes that result in carbon emissions, are referred to as being transformed by the human production activities they support, which release carbon dioxide into the atmosphere, a process that includes both direct and indirect carbon emissions. Use of Land: The processes, actions, and tactics by which land is subject to carbon emissions [6].
[Question 5]: In the calculation of carbon emissions, why is the carbon emission coefficient of construction land special?
Response: Thanks for the positive and constructive comments regarding our paper. We have rewritten this section based on your suggestions.
Detail:
Line 193-196,page 5(Revised manuscript).
“It is noteworthy to consider construction land. The carbon emissions from construction land primarily result from human production and daily life processes, and indirect estimation methods can be employed for calculation.”-> “It is worth noting that the carbon emissions of construction land mainly come from the processes of human production and life. With the acceleration of the urbanization process, the carbon emission activities on construction land have increased widely, and its carbon emission coefficient has significant differences in different years.”
[Question 6] Why is land increased in Section 3.1.
Response: We appreciate the opportunity to provide further clarification on this important point. This is an excellent suggestion. We added the reason for the change of land to the manuscript.
Detail:
Line 328-333,page 9(Revised manuscript)..
Our land use classification results show that the total land area of the Zhoushan Archi- 299
pelago increased between 2010 and 2015, mainly due to human reclamation activities. The 300
area decreased slightly from 2015 to 2020, with relatively small changes. The research area selected in this paper is the archipelago. Unlike inland cities, the total amount of land in island areas will change due to human reclamation activities and changes in coastlines. As the local government stopped the sea to land, the total amount of land gradually stabilized, but the change of the coastline still caused the total amount of land in the small area to fluctuate. View the details, the area of forest and water gradually decreased each year, while the area designated for construction steadily increased. Influenced by human reclamation activities on mudflats, grassland and mudflat areas showed an overall increase.
[Question 7]: The figure-legend text is too small.
Response: Thank you for your careful inspection. We updated all the figures in the manuscript and enlarged the size of the legend text. We have updated it in the revised manuscript.
[Question 8]: I need to make some language corrections.
Response: Thank you for your careful inspection. We tried our best to improve the manuscript and made some changes to it. These changes will not affect the content or framework of the paper. We do not list the details of all the changes here, but they are highlighted in red in the revised manuscript.
We tried our best to improve the manuscript and made some changes to the manuscript. We appreciate your earnest work and hope that the correction will meet with approval. Once again thank you very much for your comments and suggestions.
Best wishes!
Yours sincerely,
Qian Zhou
January 14, 2024
References:
- Stainforth, D.A.; Calel, R. New Priorities for Climate Science and Climate Economics in the 2020s. Nat Commun 2020, 11, 1–3, doi:10.1038/s41467-020-16624-8.
- Vale, P.M. The Changing Climate of Climate Change Economics. Ecological Economics 2016, 121, 12–19, doi:10.1016/j.ecolecon.2015.10.018.
- Liu, M.; Chen, Y.; Chen, K.; Chen, Y. Progress and Hotspots of Research on Land-Use Carbon Emissions: A Global Perspective. Sustainability 2023, 15, doi:10.3390/su15097245.